🔬 eLife

# Bidirectional fear modulation by discrete anterior insular circuits in male mice

Sanggeon Park[1,2†], Yeowool Huh[3,4†], Jeansok J Kim[5]*, Jeiwon Cho[1,2]*

[1]Department of Brain and Cognitive Sciences, Scranton College, Ewha Womans University, Seoul, Republic of Korea; [2]Brain Disease Research Institute, Ewha Brain Institute, Ewha Womans University, Seoul, Republic of Korea; [3]Department of Basic Medical Science, College of Medicine, Catholic Kwandong University, Gangneung, Republic of Korea; [4]Institute for Bio-Medical Convergence, International St. Mary's Hospital, Catholic Kwandong University, Incheon, Republic of Korea; [5]Department of Psychology, University of Washington, Seattle, United States

**Abstract** The brain's ability to appraise threats and execute appropriate defensive responses is essential for survival in a dynamic environment. Humans studies have implicated the anterior insular cortex (aIC) in subjective fear regulation and its abnormal activity in fear/anxiety disorders. However, the complex aIC connectivity patterns involved in regulating fear remain under investigated. To address this, we recorded single units in the aIC of freely moving male mice that had previously undergone auditory fear conditioning, assessed the effect of optogenetically activating specific aIC output structures in fear, and examined the organization of aIC neurons projecting to the specific structures with retrograde tracing. Single-unit recordings revealed that a balanced number of aIC pyramidal neurons' activity either positively or negatively correlated with a conditioned tone-induced freezing (fear) response. Optogenetic manipulations of aIC pyramidal neuronal activity during conditioned tone presentation altered the expression of conditioned freezing. Neural tracing showed that non-overlapping populations of aIC neurons project to the amygdala or the medial thalamus, and the pathway bidirectionally modulated conditioned fear. Specifically, optogenetic stimulation of the aIC-amygdala pathway increased conditioned freezing, while optogenetic stimulation of the aIC-medial thalamus pathway decreased it. Our findings suggest that the balance of freezing-excited and freezing-inhibited neuronal activity in the aIC and the distinct efferent circuits interact collectively to modulate fear behavior.

## eLife assessment

This work provides a **valuable** characterization of neural activity in the anterior insular cortex during fear. Using behavior, single unit recording, and optogenetic control of neural activity, the paper provides **convincing** data on the role of anterior insular circuits in bidirectionally controlling fear. The study is a great starting point on the path to testing hypotheses about bidirectional control of behavior via neural activity in anatomically defined output populations.

## Introduction

Pavlovian fear conditioning is believed to serve a crucial survival function in nature and is widely used as a preclinical model system to understand normal and abnormal fear behavior in humans (*Kim and Jung, 2006*; *Maren and Fanselow, 1996*; *Armony et al., 1998*). Functional brain imaging studies consistently implicate the insular cortex (IC) in processing fear and anxiety in humans. Specifically, healthy subjects show increased IC activity with fear conditioning (*Gottfried and Dolan, 2004*;

*For correspondence:
jeansokk@uw.edu (JJK);
jelectro21@ewha.ac.kr (JC)

†These authors contributed equally to this work

Competing interest: The authors declare that no competing interests exist.

*Klucken et al., 2009*; *Knight et al., 2009*; *Marschner et al., 2008*; *Morris and Dolan, 2004*; *Phelps et al., 2004*), while patients with anxiety disorders exhibit even greater IC activity (*Bruce et al., 2012*; *Yoon et al., 2017*; *Hoehn-Saric et al., 2004*). Animal studies also implicate the IC in fear conditioning. For instance, re-exposure of rodents to a fear-conditioned context increases IC activity as measured by c-Fos or cytochrome oxidase staining (*Beck and Fibiger, 1995*; *Bruchey and Gonzalez-Lima, 2008*). Additionally, microinjection of cobalt chloride, a non-selective synapse blocker, in the IC disrupts fear conditioning (*Alves et al., 2013*), and post-training electrolytic lesions of the IC attenuate the expression of conditioned fear (*Brunzell and Kim, 2001*).

Anatomically, the IC is divided into the anterior IC (aIC) and posterior IC (pIC) by the middle cerebral artery. Although they are reciprocally connected, the aIC and pIC have distinct connections with other brain regions (*Gogolla, 2017*; *Shi and Cassell, 1998*; *Gehrlach et al., 2020*). Specifically, the pIC receives inputs from the primary somatosensory thalamic nuclei, while the aIC receives strong inputs from the prefrontal cortex and has reciprocal connections with higher order association thalamic nuclei, such as the mediodorsal thalamus. Given its connections with cortical brain regions, the aIC is ideally positioned to integrate high-level information to regulate behaviors. Although both aIC and pIC have been implicated in fear conditioning (*Alves et al., 2013*; *Shi et al., 2020*; *de Paiva et al., 2021*; *Casanova et al., 2016*; *Park et al., 2022*), how the aIC connections to different brain regions modulate fear remains unclear.

Therefore, in this study, we investigated how the aIC neurons projecting to different brain regions process auditory fear conditioning in male mice. Using single-unit recording, we found that a subset of aIC pyramidal neurons displayed activity that was positively or negatively correlated with a conditioned freezing response, suggesting that the balance of different aIC neuronal activities regulates fear behavior. Neural tracing revealed that distinct population of aIC neurons projects to the amygdala and the medial thalamus. Optogenetic manipulations confirmed that the aIC exerts bidirectional modulation of fear behavior through the aIC-amygdala pathway (increases fear) and the aIC-medial thalamus pathway (decreases fear). Together, our study suggests that different aIC circuits are involved in bidirectional modulation of fear behavior.

## Results

### aIC activities correlate with conditioned fear

To investigate how conditioned fear is processed by individual aIC neurons, we recorded their activity during the expression of auditory fear (conditioned group) or to tone presentation (control group; *Figure 1A*). We histologically verified the recording locations to be within the aIC including the dorsal agranular insular cortex (AID), ventral agranular insular cortex (AIV), dysgranular insular cortex (DI), and granular insular cortex (GI; *Figure 1B*). Most unit signals were recorded from the AID and AIV. We only used well-isolated unit signals from the aIC for analysis (*Figure 1C*). Of the recorded neurons, we analyzed the activity of 108 putative pyramidal neurons (93% of total isolated neurons) from 11 mice, which were distinguished from putative interneurons (n=8 cells, 7% of total isolated neurons) based on the characteristics of their recorded action potentials (*Figure 1D*; see Methods for details).

As expected, the auditory fear conditioned mice significantly increased their freezing behavior during the tone conditioned stimulus (CS) presentation (*Figure 1E*, repeated-measures ANOVA, $F_{(35,210)}$ = 11.303, p<0.001), particularly during tone-on periods compared to tone-off periods (*Figure 1F* left, paired samples t-test, p<0.001). In contrast, control mice that did not undergo auditory fear conditioning showed virtually no freezing during the tone presentation (*Figure 1E and F*).

The mean of all normalized (z-scored) pyramidal neuronal activities recorded in the aIC remained steady during the experiment in both groups (*Figure 1G*). The conditioned group showed no significant correlation between the normalized firing rate and freezing behavior, nor was there any difference in the mean firing rate between the tone-on and tone-off periods (*Figure 1H*). Similar trends were observed in the normalized neuronal activity of control mice (*Figure 1G and H*). These results suggest that the mean activity of all recorded aIC pyramidal neurons remains relatively constant, regardless of the fear experience.

Several studies report reduction in theta frequency in presence of the CS in several brain regions including the hippocampus, medial prefrontal cortex, and amygdala (*Moita et al., 2003*; *Lesting et al., 2011*; *Buzsáki, 2002*). Similarly, the power of aIC local field potential in the <50 Hz frequency,

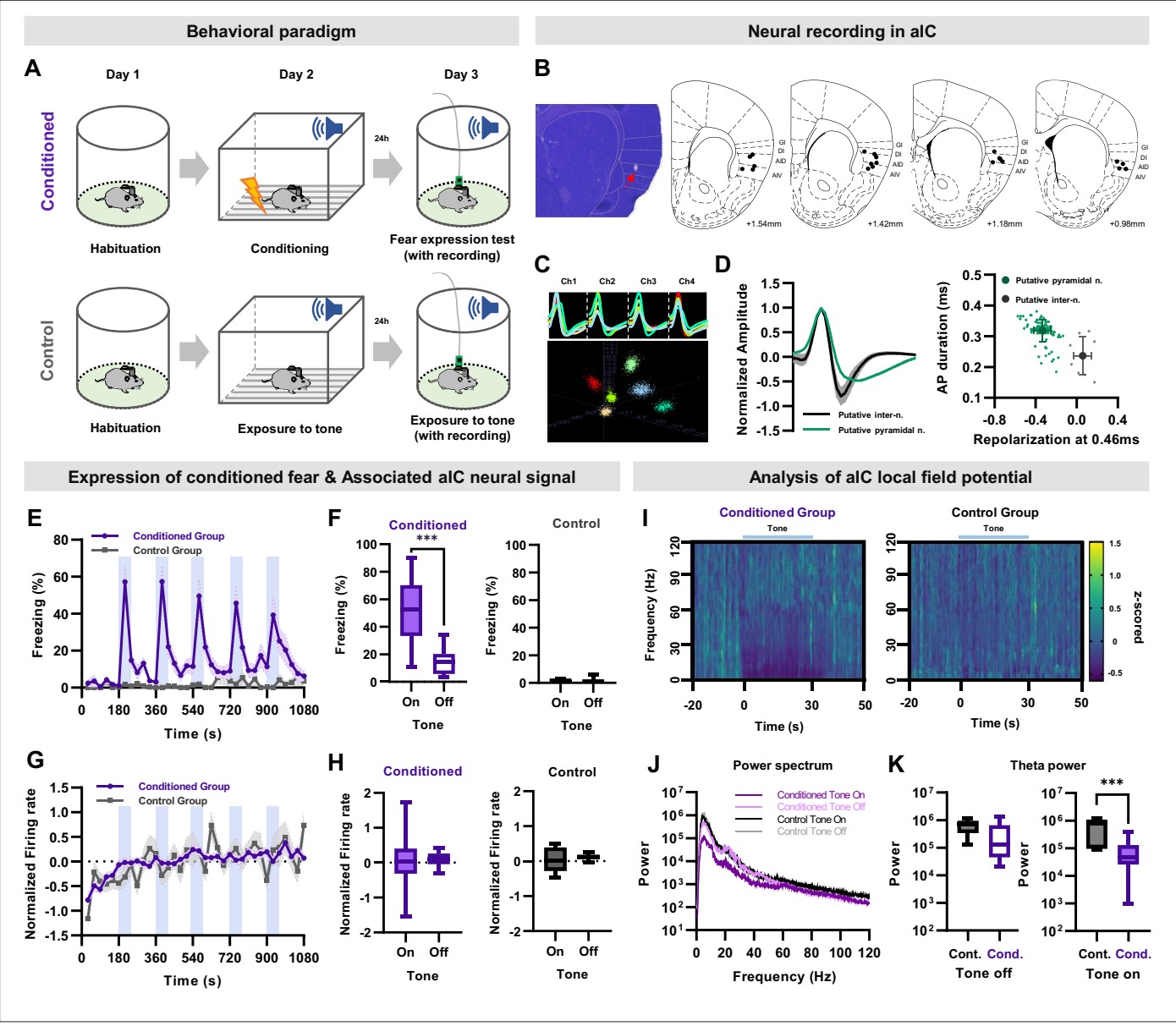

**Figure 1.** Single unit recording of aIC neurons during conditioned fear expression. (**A**) Schematic of the auditory fear conditioning and single unit recording procedures. Neuronal activities and behaviors were simultaneously recorded during the expression of conditioned fear. The control group underwent the same procedures except for fear conditioning. (**B**) Histological sample showing the tip of a recording electrode, indicated with a red arrow. Brain atlas maps indicating the locations of aIC recording sites used for analysis. The black dots represent the recording site. GI: granular insular cortex, DI: dysgranular insular cortex, AID: dorsal agranular insular cortex, AIV: ventral agranular insular cortex. (**C**) Sample cluster cutting of neuronal recording acquired from a tetrode. (**D**) Juxtaposed action potential shape of all recorded neurons (left). Putative pyramidal neurons and interneurons were distinguished based on the shape of the action potential. The plot on the right shows action potential duration and repolarization. Putative pyramidal neurons are represented by green dots, while putative interneurons are represented by gray dots. (**E**) Mean (± SEM) freezing (fear) behavior of conditioned or control mice during five tone presentations (shaded blue bars). Conditioned N=11, Control N=3 mice. (**F**) Freezing of conditioned (left) or control (right) mice during five tone-on and tone-off sessions. (**G**) Mean (± SEM) normalized (z-scored) firing rate of aIC pyramidal neurons simultaneously recorded with freezing (shaded blue bars represent tone presentations). Conditioned N=108 putative pyramidal neurons from 11 mice, Control N=14 putative pyramidal neurons from 3 mice. (**H**) Firing rate of conditioned (left) or control (right) mice during five tone-on and tone-off sessions. (**F and H**) Paired sample t-test was used to compare means between tone-on and tone-off sessions in each group . *** p<0.001. (**I**) Spectrogram of aIC local field potential (LFP) recording of the conditioned and control groups before and during tone presentation. Conditioned N=21 LFP recordings from 11 mice, Control N=7 LFP recordings from 3 mice. (**J**) Power spectrum of aIC local field potential of the conditioned and control groups. Conditioned N=21 LFP recordings from 11 mice, Control N=7 LFP recordings from 3 mice. (**K**) Theta power analysis before tone (left) and during tone (right) of the conditioned and control groups. Cond.: conditioned group, Cont.: control group. The Mann–Whitney U test was used to assess the statistical difference between the conditioned and control groups in each session. *** p<0.001.

which includes the theta frequency, tended to be reduced in the conditioned group compared to the control during CS (*Figure 1I and J*). Quantitative analysis revealed that theta frequency (6 Hz) was significantly reduced during, but not before, CS presentation in the conditioned group compared to the control (*Figure 1K*).

To examine the relationship between firing rates of individual neurons and conditioned freezing behavior in detail, we attempted to identify a population of neurons whose activity changed in response to the conditioned tone stimulus. We identified three types of aIC neuronal responses: neurons that decreased firing rate (inhibited cell), neurons that increased firing rate (excited cell), and neurons that did not change their firing rate (no-change cell) at the presence of CS (*Figure 2A*). Activity changes of the inhibited and excited cell types showed a tendency to be correlated with freezing behavior (*Figure 2A*, bottom row). To further analyze the relationship between neuronal activity changes and freezing, we used the Pearson's correlation analysis and found three types of aIC neuronal responses: neurons whose firing rate increased in parallel with the freezing response ('freezing-excited cells'), neurons whose firing rate decreased with freezing ('freezing-inhibited cells'), and neurons that had no significant changes correlated with freezing ('non-responsive cells'). The distribution of all coefficients is shown in *Figure 2B*. Among the neurons recorded from fear conditioned mice, 29.6% were responsive (n=32/108) and 70.4% were non-responsive (n=76/108; *Figure 2A*). Among the responsive cells, there were proportionally more freezing-excite cells (n=21/32, 65.6%) than freezing-inhibited cells (n=11/32, 34.4%), but the trend was not statistically significant (binomial proportions test, p=0.11; *Figure 2B*).

Histological analysis of the recording locations of respective neuronal-response types suggested that the freezing-excited and freezing-inhibited cells may be located in different layers of the aIC (*Figure 2C*). The freezing-excited cells tended to be recorded in the middle and deep layers (L5 and 6; *Figure 2C*, red dots), while the freezing-inhibited cells tended to be recorded in the superficial and middle layers (L2/3, 4, and 5; *Figure 2C*, blue dots). Tetrodes that recorded both freezing-excited and freezing-inhibited cells tended to be located in the middle layer (L5; *Figure 2C*, red and blue gradation dot). The non-responsive cells were recorded from all layers of the aIC. To visualize any differences in firing rate changes among different neuronal-response types in fear conditioned and control mice, the mean normalized firing rate of individual aIC neurons was sorted in ascending order of their respective Pearson's correlation coefficients (*Figure 2D*). Freezing-inhibited cells showed relatively higher tone-off firing rates, which decreased during tone presentation (*Figure 2D*, conditioned upper). Freezing-excited cells, in contrast, showed a lower tone-off firing rate that increased during tone presentation (*Figure 2D*, conditioned lower). The non-responsive cells, interestingly, showed an intermediate tone-off firing rate between those of the freezing-inhibited and freezing-excited cells and remained relatively constant throughout the recording (*Figure 2D*, conditioned middle). The neuronal activity of control mice was similar to the non-responsive cells of the fear conditioned mice (*Figure 2D*, control). The differences in baseline firing rate before CS presentation of different cell-response types are also reflected in raw, not normalized, values (*Figure 2E*; one-way ANOVA, $F=6.65$, $p<0.001$). These results suggest that subpopulations of pyramidal neurons in the aIC may become primed to respond to the fear-conditioned stimulus.

Pearson's correlation coefficient between mean normalized firing rate and freezing of each neuronal response type revealed that the activities of the freezing-excited and freezing-inhibited neurons were significantly correlated with freezing behavior in opposite directions (*Figure 2F*; freezing-excited cell $r=0.81$, $p<0.05$; freezing-inhibited cell $r=-0.68$, $p<0.05$). As expected, no correlation was found for the non-responsive neurons ($r=0.13$, $p=0.44$). For detailed analysis of each neuronal response type's activity changes that occurs during CS, we compared the values of freezing-inhibited and freezing-excited cells in smaller time bins (1 s). The tone-induced firing rate peaks of the freezing-inhibited and freezing-excited cells were significantly different (*Figure 2G*, Mann-Whitney U Test, $p<0.05$); the freezing-inhibited cells peaked at 10.5 s, while the freezing-excited cells peaked at 17.6 s. We also analyzed additional qualities of the aIC neurons, such as the presence of neurons that precisely signaled the beginning or termination of the tone, the tone-onset and tone-offset cells, respectively. However, none of the recorded aIC neurons, neither the conditioned group's nor the control's, were tone-onset or tone-offset cells (*Figure 2H*).

We then assessed the correlation between neuronal activity and tone. None of the neuronal activity of control mice (n=14 cells, 3 mice) significantly correlated with the tone ($r=0.027$, $p=0.87$),

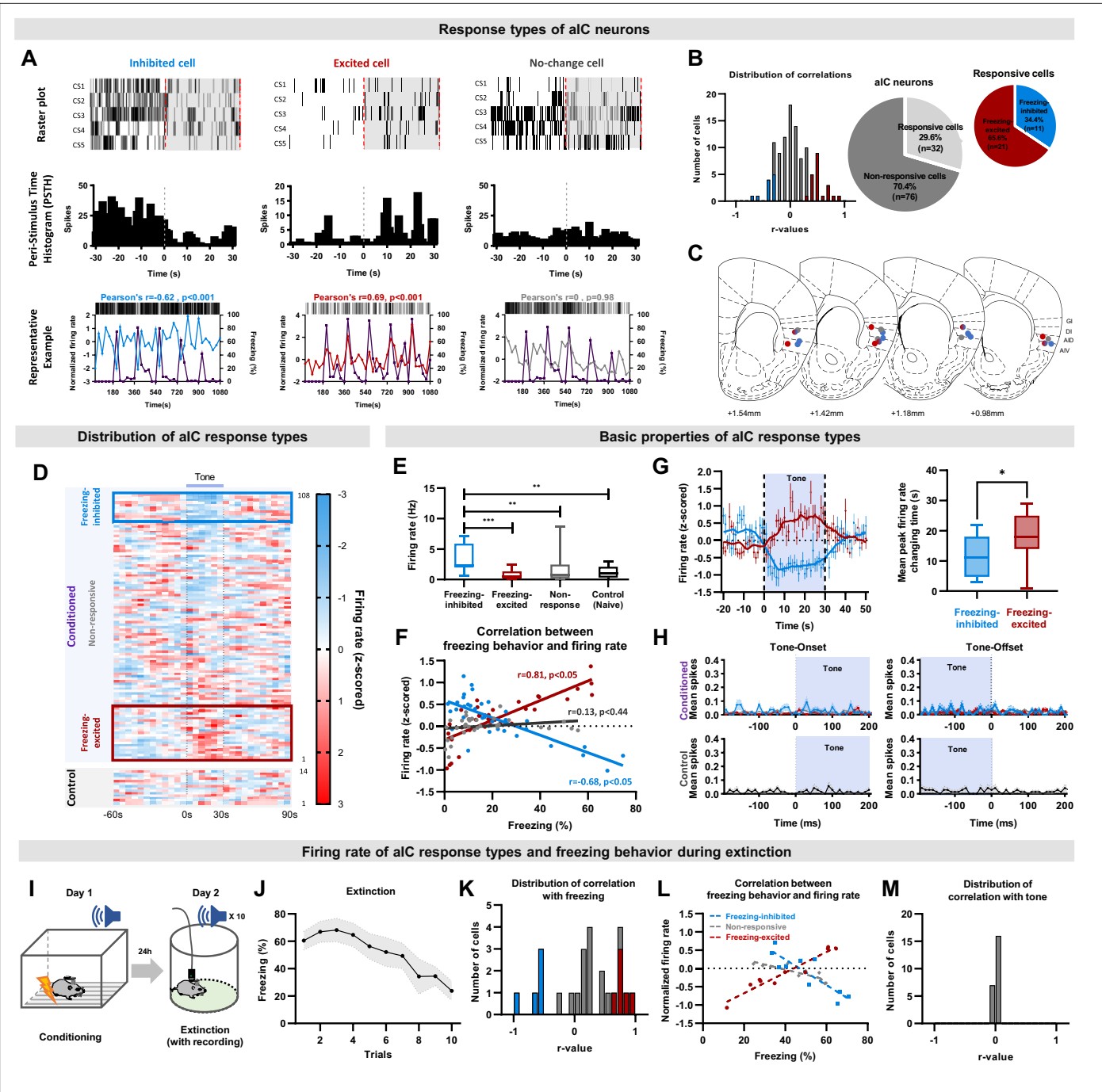

**Figure 2.** The activity of aIC pyramidal neurons bidirectionally correlates with freezing (fear) behaviors. (**A**) Example of aIC neurons that decreased (inhibited-cell, left), increased (excited-cell, center), or did not change (no-change cell, right) their firing rate over five CS presentations (CS1-5). Top row: raster plot, each tick mark indicates when an action potential occurred. Middle row: peri-stimulus time histogram of each cell type over five CS presentations. Bottom row: plot of neuronal activity changes and freezing behavior of each cell type. Blue line: activity of inhibited cell, Red line: activity of excited cell, Gray line: activity of no-change cell, Purple line: freezing behavior. (**B**) Distribution of Pearson's correlation coefficient values between freezing behavior and aIC neuronal activities of the fear conditioned group. The red bars represent cells with significant positive correlation (freezing-excited cells), blue bars represent cells with significant negative correlation (freezing-inhibited cells), and gray bars represent cells with no significant correlation (non-responsive) with freezing behavior. Significance was determined at p<0.05. Relative ratio of the non-responsive and responsive (freezing-excited or freezing-inhibited) cell types recorded in the fear conditioned group (right pie chart). (**C**) Locations of cell-types recorded. Red dot: freezing-excited or freezing-excited/non-responsive cells, blue dot: freezing-inhibited or freezing-inhibited/non-responsive cell, red and blue gradation dot: combination of freezing-excited/freezing-inhibited/non-responsive cells, gray dot: non-responsive cells. (**D**) Distribution of all recorded aIC neuronal activities (z-scored) before, during, and after CS (tone). presentation. Each line represents the mean response of one cell to five CSs, aligned from the

*Figure 2 continued on next page*

*Figure 2 continued*

lowest to the highest firing rate change in response to CS. The blue box represents cells classified as freezing-inhibited cells, and the red box represents cells classified as freezing-excited cells. (**E**) Baseline firing rates of freezing-excited cells (n=21), freezing-inhibited cells (n=11), non-responsive cells (n=76), and control cells (n=14). The Mann-Whitney U test was used to assess statistical differences between responsive types (** p<0.01, *** p<0.001). (**F**) Pearson correlation analysis for each neuronal response type and freezing behavior. (**G**) Firing rate changes of freezing-inhibited and freezing-excited cells analyzed in 1 s intervals (left). The box plot shows the difference in time that each cell type takes to reach peak changes. The Mann–Whitney U test was used to assess the statistical difference between the time that freezing-inhibited and freezing-excited cells take to reach peak firing rate change. * p<0.05 (**H**) Tone onset and offset analysis of the conditioned and the control groups. Blue line: freezing-inhibited, red line: freezing-excited, gray line: non-responsive cells. Mean (± standard deviation) spikes per second. (**I**) Schematic drawing of the conditioned fear extinction procedure. (**J**) Freezing (fear) behavior of mice during conditioned fear extinction (ten tone presentations, N=7 mice). Data are presented as Mean (± SEM). (**K**) Distribution of Pearson's correlation coefficient values between freezing behavior and aIC neuronal activities. The red bars represent cells with significant positive correlation (freezing-excited cells, n=5 cells), blue bars represent cells with significant negative correlation (freezing-inhibited cells, n=5 cells), and gray bars represent cells with no significant correlation (n=25 cells) with freezing behavior. Significance was determined at p<0.05. (**L**) Pearson correlation analysis for each neuronal response type and freezing behavior. (**M**) Distribution of Pearson's correlation coefficient values between tone and aIC neuronal activities.

but a subset of aIC neurons in conditioned mice showed significant correlation with tone (freezing-excited cells, $r$=0.44, p<0.05, n=9; freezing-inhibited cells, $r$=−0.51, p<0.05, n=5). During the five-tone presentation, however, freezing behavior was still high, and therefore difficult to distinguish whether aIC neuronal activities correlate with freezing, tone, or both. To differentiate what the aIC neuronal activities represents, we carried out a fear extinction experiment (*Figure 2I*) where conditioned freezing behavior reduces gradually over the ten trials (*Figure 2J*). Just like the five-tone presentation experiment, we found the freezing-excited and freezing-inhibited cells that correlated with freezing (*Figure 2K*). Pearson's correlation coefficient between mean normalized firing rate and freezing show that the activities of the freezing-excited and freezing-inhibited cells were significantly correlated with freezing behavior in opposite directions (*Figure 2L*; freezing-excited cells, $r$=0.81, p<0.05; freezing-inhibited cells, $r$=−0.68, p<0.05). Interestingly, none of the recorded cells correlated with tone (*Figure 2M*). Together, these results suggest that the freezing-excited and freezing-inhibited cells found in the aIC of fear-conditioned mice are highly correlated with freezing behavior, not tone.

## Manipulating aIC activities modulate conditioned fear behavior

Our results suggest that the emergence of freezing-excited and freezing-inhibited cells following fear conditioning may be critical in regulating freezing (fear) behavior. To investigate this further, we tested whether artificially increasing or decreasing the activity of aIC pyramidal neurons with optogenetics could affect freezing behavior. Viral vectors were used to express channelrhodopsin-2 (ChR2: AAV-CaMKII-ChR2-eYFP), halorhodopsin (NpHR3: AAV-CaMKII-NpHR3-eYFP), or fluorophore (control: AAV-CaMKII-eYFP) in the aIC pyramidal neurons and we assessed how optical stimulation affects conditioned fear behavior in each group (*Figure 3A*). We only analyzed the behavior of mice with confirmed viral expression in the aIC (*Figure 3B*). Viral expression patterns did not differ among groups (*Figure 3—figure supplement 1*). In a separate group of mice, we assessed whether optical stimulation induces aberrant behaviors or affects locomotion with the open-field test. Neither activation of ChR2 nor NpHR3 triggered any aberrant behaviors, and locomotion did not differ among groups (*Figure 3C*). Optical stimulation of ChR2 or NpHR3 specially activated or inhibited the activity of aIC pyramidal neurons, respectively (*Figure 3D and E*). ChR2 stimulation increased brain rhythms in the 20 Hz and 40 Hz range (*Figure 3D*, spectrogram) because light stimulation was delivered at 20 Hz pulses. Continuous light was delivered for NpHR3 stimulation, so it did not induce specific frequency modulation (*Figure 3E*, spectrogram). Simultaneous local field potential (LFP) and single unit analysis before, during, and after optical stimulations confirmed that optostimulation of the aIC neurons did not induce aberrant seizure-like synchronized activities.

After confirming that optical activation parameters used in the study does not interfere with locomotion, we examined whether optogenetically manipulating the activity of aIC pyramidal neurons modulates conditioned freezing behavior. As predicted, optogenetic excitation or inhibition of aIC pyramidal neurons in the right hemisphere produced distinct changes in freezing behavior (*Figure 3F*). During a 30 s tone +optostimulation, there was no difference in freezing behavior among groups (*Figure 3G*; Kruskal-Wallis, $H$(2) = 2.489, p=0.288). However, during the tone-off sessions, freezing behavior among groups differed significantly (*Figure 3G*; Kruskal-Wallis, $H$(2) = 22.937, p=0.000;

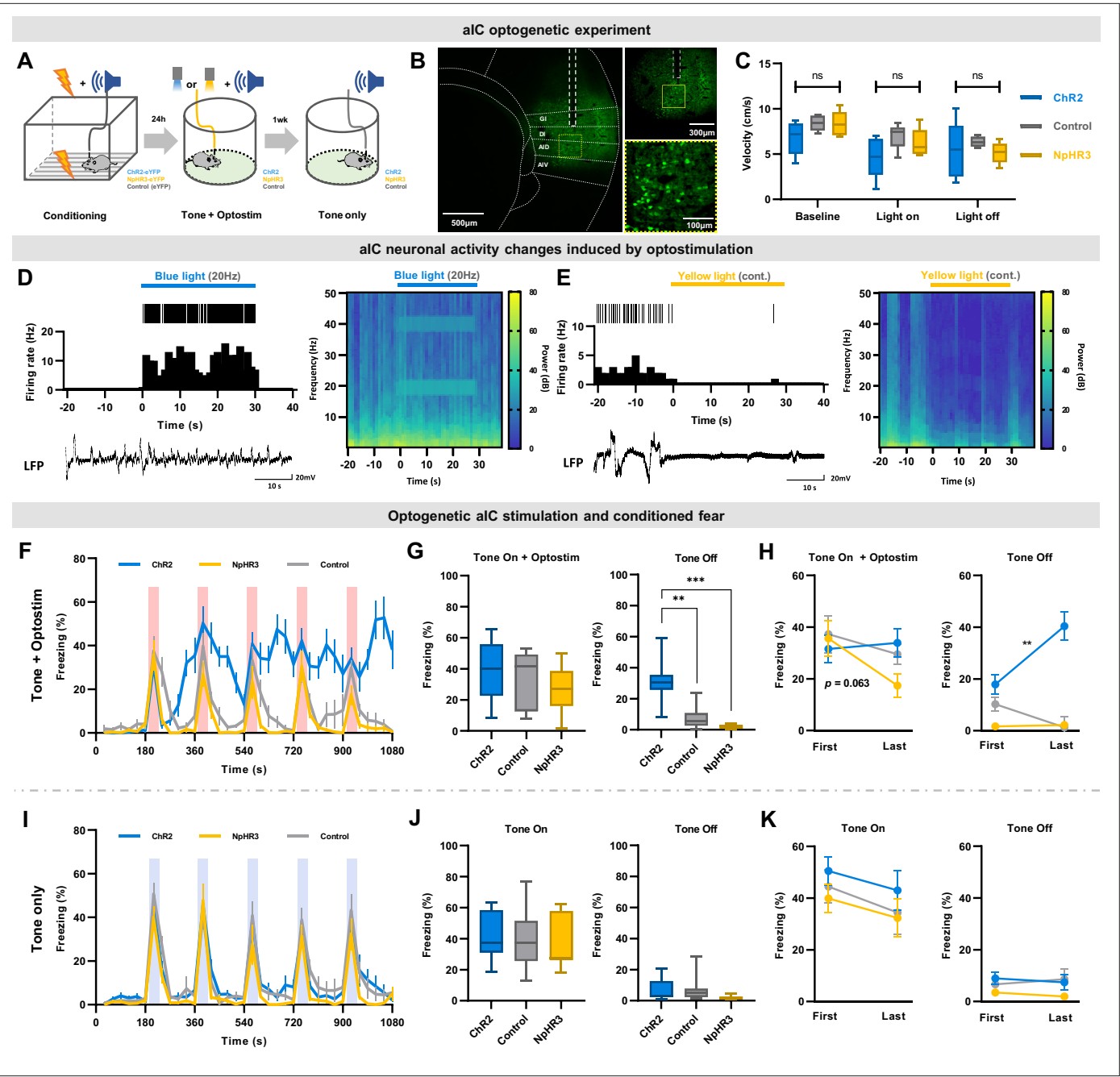

**Figure 3.** Optogenetic regulation of aIC pyramidal neuron output modulates conditioned fear expression. (**A**) Schematic drawing of the experimental protocol. Experimental groups consisted of viral vectors injected for excitation (AAV-CaMKII-ChR2-eYFP), inhibition (AAV-CaMKII-NpHR3-eYFP), and control (AAV-CaMKII-eYFP). (**B**) Histological sample showing viral expression pattern and optic fiber placement. Dashed white line depicts location of optic fiber placement. Yellow dotted square indicates the ×40 magnified region shown below. (**C**) Velocity before (baseline), during (light on), and after (light off) optostimulation of the three groups in an open field test (ChR2=13 mice, NpHR3=7 mice, Control=13 mice). (**D**) Simultaneous single unit and local field potential (LFP) recording during the 20 Hz blue-light activation of ChR2-expressing neurons. Juxtaposed unit activity (each tick indicates an action potential), peri-stimulation time histogram, and LFP (left). Local brain rhythm (frequency) changes induced by blue light stimulation (right). (**E**) Simultaneous single unit and local field potential (LFP) recording during the continuous (cont.) yellow-light activation of NpHR-expressing neurons. Juxtaposed unit activity (each tick indicates an action potential), peri-stimulation time histogram, and LFP (left). Local brain rhythm (frequency) changes induced by yellow light stimulation (right). (**F**) Mean (± SEM) freezing behavior of mice during the five CS (tone) and optostimulation delivery. (**G**) Freezing behavior of the five tone-on with optostimulation and tone-off periods. (Kruskal–Wallis test; **p<0.01, ***p<0.001). (**H**) Comparison between the first and the last tone +optostimulation and tone-off session freezing behavior among groups (Wilcoxon signed ranks test; **p<0.01). (**I**) Mean (± SEM) freezing behavior of mice during five CS-only presentations without any optostimulation. (**J**) Freezing behavior of the five tone-on and

*Figure 3 continued on next page*

Figure 3 continued

tone-off periods. (**K**) Comparison between the first and the last tone and tone-off session freezing behavior among groups. (**F–K**) Blue line represents optical excitation (ChR2, N=13 mice), yellow line represents optical inhibition (NpHR3, N=7 mice), and gray line represents control (Control, N=13 mice). (**F, G, L, J**) The Kruskal–Wallis test was used to test statistical significance of among groups. (**H and K**) The Wilcoxon signed ranks test was used to compare means of freezing behavior between first and last sessions in each group.

The online version of this article includes the following figure supplement(s) for figure 3:

**Figure supplement 1.** Overlaid viral expression patterns of the excitation (ChR2), control (eYFP), and inhibition (NpHR3) groups in the aIC.

post hoc Mann-Whitney, ChR2 vs. control p=0.000; NpHR vs. control p=0.019). Conditioned freezing responses of the ChR2 group were sustained even after tone +optostimulation ended, and tended to accumulate with repeated stimulations, particularly during the tone-off sessions. This was distinct from the conditioned freezing responses observed in the NpHR3 or control groups (*Figure 3G*). In the NpHR3 group, freezing immediately decreased after tone +optostimulation ended (*Figure 3G*). When comparing the freezing behavior of each group during their first and last tone +optostimulation trials, inhibition of aIC pyramidal neurons (NpHR3) reduced the freezing level between the first and the last tone +optostimulation (*Figure 3H*, left; Wilcoxon signed test, p=0.063). During the tone-off period, the ChR2 group significantly increased freezing after the last stimulation compared to the first (*Figure 3H*, right; Wilcoxon signed test, **p<0.01).

These group differences in fear expression were absent in the experiment carried out without optostimulation (ANOVA group effect: $F_{(2,30)}$ = 1.076, p=0.354; *Figure 3I and J*), confirming that group differences in fear expression were induced by the activation or inhibition of pyramidal neurons in the aIC. Likewise, the accumulative influence in behavior was absent without optostimulation (*Figure 3K*).

## Distinct aIC circuits bilaterally control conditioned fear

Despite optogenetic modulation of aIC pyramidal neurons affecting fear behavior, the effect was unclear, as there were no differences in freezing behavior during optical stimulation delivery among groups. One possible explanation for this is that the aIC circuits that regulate fear behavior in opposing directions were simultaneously activated to cancel out the effects. To investigate further, we looked into aIC projections to various brain areas that could potentially influence fear behavior. The aIC projects to the amygdala (BLA, basolateral amygdala; and CeA, central amygdala) and the medial thalamus (MD, mediodorsal thalamus; and CM, centromedial thalamus; *Gehrlach et al., 2020*). Of the many brain regions the aIC project to, we focused on these brain areas due to their known involvement in fear regulation and the lack of reciprocal connections between them and the pIC.

To investigate whether the same or different population of neurons in the aIC project to the medial thalamus and the amygdala, we injected retrograde tracers in respective regions (*Figure 4A*). Two different colors of fluorophore-conjugated Cholera toxin subunit B were injected into the medial dorsal thalamus (Alexa Fluor 594, red) and the amygdala (Alexa Fluor 488, green) in each mouse (*Figure 4B and C*). A week later, fluorophore expression was examined in the aIC region. Tracer injection was restricted to the medial dorsal thalamus and the amygdala (*Figure 4A–C* and *Figure 4— figure supplement 1*). Surprisingly, none of the neurons in the aIC expressed both green and red fluorophores, suggesting that non-overlapping populations of aIC neurons project to the medial dorsal thalamus and the amygdala (*Figure 4D*). The neurons projecting to the medial dorsal thalamus and the amygdala were also located in different layers within the aIC; the aIC neurons projecting to the medial dorsal thalamus appeared mainly in the deep layers (approximately layer 6), while the aIC neurons projecting to the amygdala appeared in a relatively more superficial layer (approximately layer 5). Few aIC→amygdala neurons were present in layer 6, but we found no aIC→medial thalamus neurons in layer 5. The number of aIC neurons projecting to the medial dorsal thalamus was greater than those projecting to the amygdala even though they were populated in a more restricted area (*Figure 4E*). Although the recording location of the freezing-excited and freezing-inhibited cells does not exactly match the tracing results, freezing-excited cells were recorded from deep layers while freezing-inhibited cells were recorded from middle and superficial layers (*Figure 2B*). Collectively, data suggest that the aIC→medial thalamus neurons could be freezing-excited cells, while the aIC→amygdala neurons may be freezing-inhibited cells.

To distinguish the roles of the aIC→medial thalamus and the aIC→amygdala projections, we performed bilateral optical stimulation of the respective aIC terminals expressing ChR2 during the

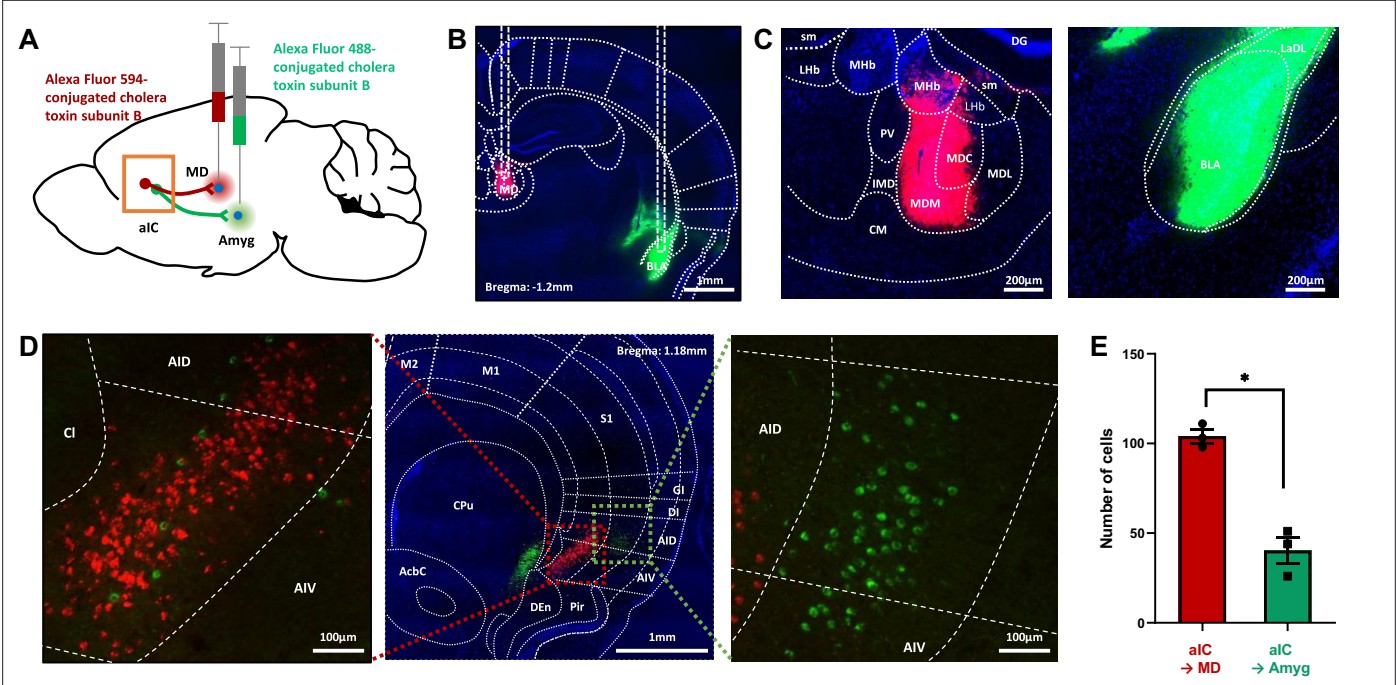

**Figure 4.** Tracing of aIC neurons that projects to the thalamus or the amygdala. (**A**) Schematic drawing of cholera toxin B (CTB) retrograde tracers injected into the amygdala (green) and thalamus (red). (**B**) Image showing the injection sites. (**C**) High-resolution of the panel b image. sm: stria medullaris of the thalamus, LHb: lateral habenular nucleus, MHb: medial habenular nucleus, DG: dentate gyrus, PV: paraventricular thalamic nucleus, IMD: intermediodorsal thalamic nucleus, MDM: mediodorsal thalamic nucleus, medial part, MDC: mediodorsal thalamic nucleus, central part, MDL: mediodorsal thalamic nucleus, lateral part, CM: central medial thalamic nucleus, LaDL: lateral amygdaloid nucleus, dorsolateral part, and BLA: basolateral amygdaloid nucleus, anterior part. (**D**) Sample image of neurons in the aIC that either project to the thalamus (red) or the amygdala (green). A high-resolution image of the thalamus-projecting aIC neurons is shown on the left, and that of the amygdala-projecting aIC neurons is shown on the right. M2: secondary motor cortex, M1: primary motor cortex, S1: primary somatosensory cortex, GI: granular insular cortex, DI: dysgranular insular cortex, AID: agranular insular cortex, dorsal part, AIV: agranular insular cortex, ventral part, Pir: piriform cortex, Den: dorsal endopiriform nucleus, CPu: caudate putamen (striatum), and AcbC: accumbens nucleus, core. (**E**) The number of aIC neurons that were labeled to either project to the thalamus (Alexa fluor 594, n=3 sections per mice, N=3 mice) or the amygdala (Alexa fluor 488, n=3 sections per mice, N=3 mice). Data are presented as mean ± SEM. The Mann-Whitney U test was used to assess statistical difference (* p<0.05).

The online version of this article includes the following figure supplement(s) for figure 4:

**Figure supplement 1.** Overlaid tracer expression patterns of all samples at the injection sites (MD: mediodorsal thalamus and BLA: basolateral amygdala).

expression of conditioned fear (***Figure 5A and E***). The area of viral expression in the aIC were similar among groups (***Figure 5—figure supplement 1***). As shown in the sample expression patterns, the aIC projection to the thalamus terminated mainly in the MD and CM regions (***Figure 5B***), while the aIC projection to the amygdala terminated mainly in the BLA and CeA (***Figure 5F***). Interestingly, activation of the aIC→medial thalamus and the aIC→amygdala projections induced completely different effects. While activating the aIC→medial thalamus projection significantly reduced freezing compared to the control (***Figure 5C***, blue line; repeated measures ANOVA, $F_{(1, 17)}$=10.666, *** p<0.001), activating the aIC→amygdala projection significantly enhanced freezing compared to the control (***Figure 5G***, red line; repeated measures ANOVA, $F_{(1, 12)}$=23.663, *** p<0.001). Also, when opto-stimulation was turned off, and only the conditioned tone was delivered (retrieval test), there were no differences in the freezing level between groups (***Figure 5C and G***). The results indicate that the behavioral differences were indeed induced by the activation of respective projections. In addition, we compared freezing levels of tone +optostim and tone-off periods to assess whether activation of these pathways produces a lasting effect on behavior. Unlike the phenomenon observed in aIC activation with ChR2 (***Figure 3F***), activation of neither the aIC→medial thalamus nor the aIC→amygdala projection produced any lasting behavioral effects between light stimulations (***Figure 5D and H***). The tracing and optogenetics manipulation results support the hypothesis that distinct hardwired circuits of the

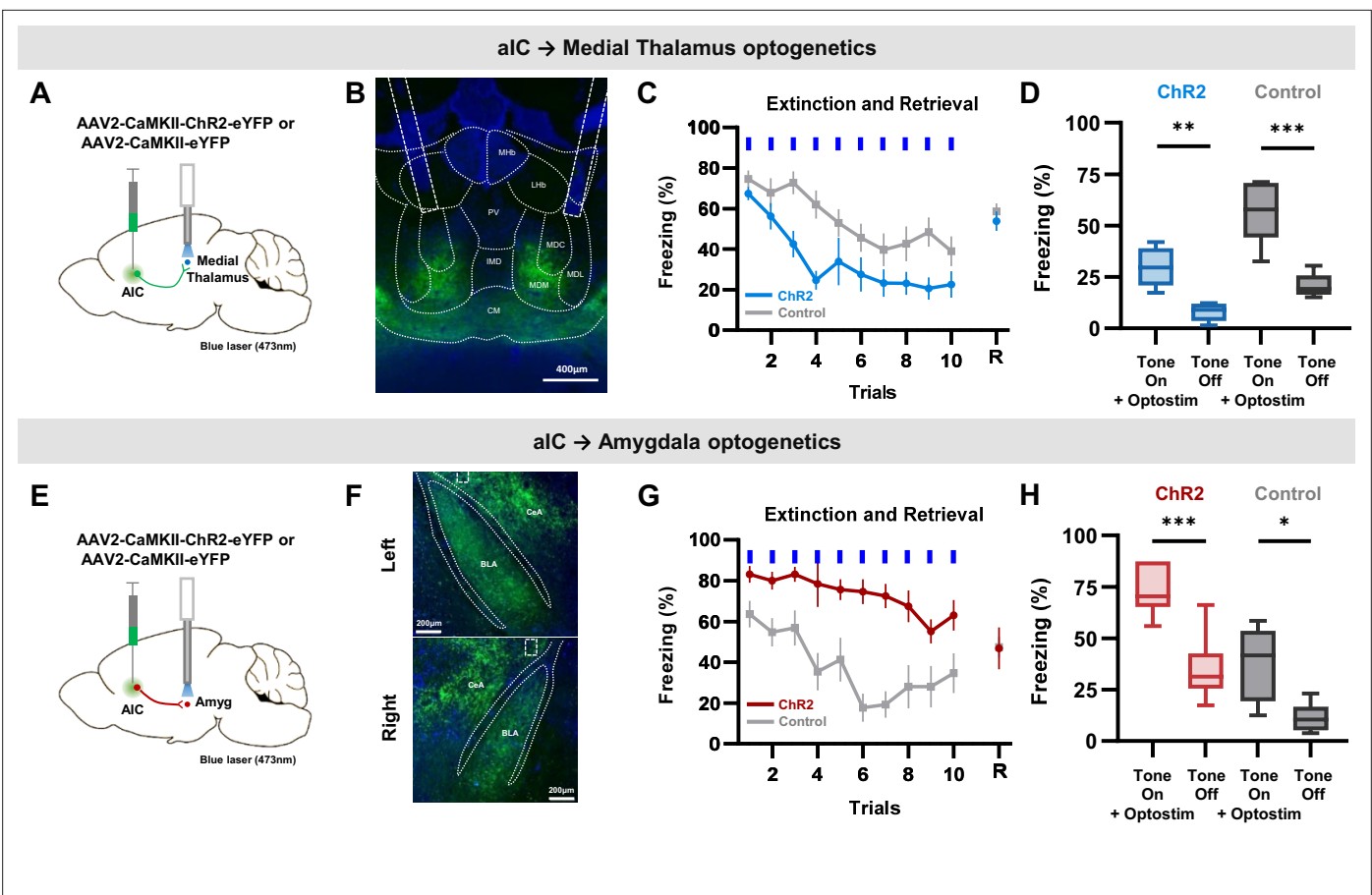

**Figure 5.** Specific optogenetic activation of aIC projection to the medial thalamus or the amygdala bidirectionally regulate conditioned fear expression.
(**A**) Schematic drawing of the adeno-associated virus (AAV) injection and optostimulation sites. Viral vectors for opsin expression (AAV-CaMKII-ChR2-eYFP) or control (AAV-CaMKII-eYFP) were bilaterally injected in the aIC, and bilateral optostimulation was delivered in the mid-thalamic region.
(**B**) Example expression pattern of aIC projection terminals in the thalamic region and the location of bilateral optic fiber placement outlined by dashed lines. LHb: lateral habenular nucleus, MHb: medial habenular nucleus, PV: paraventricular thalamic nucleus, IMD: intermediodorsal thalamic nucleus, MDM: mediodorsal thalamic nucleus, medial part, MDC: mediodorsal thalamic. nucleus, central part, MDL: mediodorsal thalamic nucleus, lateral part, CM: central medial thalamic nucleus. (**C**) Mean (± SEM) freezing behavior of mice expressing ChR2 (N=12) or control (eYFP, N=8) induced by optostimulation of aIC projection terminating in the thalamus to repeated CS (tone). Blue bars indicate when tone and optostimulation were delivered. Last point indicates freezing behavior when tone was delivered without optostimulation 24 hr after the extinction protocol (retrieval). (**D**) Freezing behavior during the 'Tone on +optostim' and 'Tone off' of the aIC thalamus ChR2 and control groups. (**E**) Schematic drawing of AAV injection and optostimulation sites. Viral vectors for opsin expression (AAV-CaMKII-ChR2-eYFP) or control (AAV-CaMKII-eYFP) were bilaterally injected in the aIC, and bilateral optostimulation was delivered in the anterior-amygdala region. (**F**) Example expression pattern of aIC projection terminals in the amygdala and location of bilateral optic fiber placement outlined by dashed lines. CeA: central nucleus of the amygdala, BLA: basolateral amygdaloid nucleus, anterior part. (**G**) Mean (± SEM) freezing behaviors of mice expressing ChR2 (N=6) or control (eYFP, N=6) induced by optostimulation of aIC projection terminating in the amygdala to repeated CS (tone). Blue bars indicate when tone and optostimulation were delivered. Last point indicates freezing behavior when tone was delivered without optostimulation 24 hr after the extinction protocol (retrieval). (**H**) Freezing behavior during the 'Tone on +optostim' and 'Tone off' of the aIC-amygdala ChR2 and control groups. (**C and G**) Two-way repeated measure ANOVAs were performed to compare freezing behavior between groups. (**D and H**) Two-way ANOVA with Bonferroni post hoc was used to compare the effect of 'Tone on +optostim' and 'Tone off' as well as the difference between the ChR2 and control groups. There was a statistically significant group effect (ChR2 vs control; Medial Thalamus: p<0.001, Amygdala: p<0.001) and light effect (Tone on +optostim vs Tone off; Medial Thalamus: p<0.001, Amygdala: p<0.001), but no interaction effect between group and light (Medial Thalamus: p=0.128, Amygdala: p=0.311). Paired t-test was used to compare means of 'Tone on +optostim' and 'Tone off' in each group. (*p<0.05, **p<0.01, ***p<0.001).

The online version of this article includes the following figure supplement(s) for figure 5:

**Figure supplement 1.** Overlaid opsin expression patterns in the anterior insular cortex of the different experimental groups.

aIC, one projecting to the amygdala and the others projecting to the medial thalamus, regulate conditioned fear bidirectionally.

## Discussion

Our study shows that neuronal activity in the aIC can regulate conditioned fear behavior bidirectionally via at least two different subpopulations of neurons that project either to the amygdala or the thalamus. Specifically, we found two types of aIC pyramidal neurons whose activity positively or negatively correlates with conditioned fear behavior, consistent with our previous finding that restraint stress-induced defensive behaviors (*Park et al., 2022*) correlate with the activity of aIC pyramidal neurons. While previous studies have reported excitatory or inhibitory neuronal responses to conditioned stimuli in different conditioning paradigms and brain areas, including the pIC, this study found that the ratio of these neurons differed in the aIC. In the pIC, a significantly larger population of excitatory neurons was reported compared to inhibitory neurons (*Casanova et al., 2018*), but in the aIC, we found approximately equal number of freezing-excited and freezing-inhibited neurons. Difference in this ratio may partially contribute to the differential roles of the aIC and pIC in processing fear. The brain's balance between excitation and inhibition (E/I) has been proposed to play a crucial role in its normal functioning. An E/I imbalance has been linked to various disorders, such as epilepsy, schizophrenia, and autism spectrum disorder (*Yizhar et al., 2011*). Group activity of neurons in the pIC measured with fiberphotometry, interestingly, exhibited fear state dependent activity changes— decreased activity with high fear behavior and increased activity with lower fear behavior (*Klein et al., 2021*)—suggesting that group activity of the pIC may be involved in maintaining appropriate level of fear behavior. This study has not investigated whether the aIC also show fear state dependent changes in group neuronal activities; however, approximately equal number of neurons were freezing-excited and freezing-inhibited neuron found in our study suggests that the aIC maintains E/I balance at the cellular activity level when processing fear, possibly by activating different circuits.

While both the freezing-excited and freezing-inhibited aIC cells showed significant correlations with freezing behavior, their temporal firing rate changes during CS presentation were different. Specifically, the activity of the freezing-inhibited cells dropped before the freezing-excited cells increased in response to the CS presentation. Previous studies have demonstrated that temporal differences in the arrival of excitatory and inhibitory inputs can create a time window for local signals to be transmitted while maintaining the global E/I balance (*Vogels and Abbott, 2009*; *Bhatia et al., 2019*). We hypothesize that the observed differences in the temporal activity of the freezing-excited and freezing-inhibited aIC cells may provide such a window for orchestrating signal gating and processing. This suggests that the aIC may play a critical role in regulating the balance between excitation and inhibition during fear conditioning.

Another distinction between the aIC and pIC may be related with anxiety, as a recent study showed that group activity of aIC neurons, but not that of the pIC, increased when mice explored anxiogenic space (open arms in an elevated plus maze, center of an open field box) (*Nicolas et al., 2023*). Our previous study also showed that the aIC is involved in stress-induced anxiety and that neuronal activity correlated with stress-induced defensive behavior (burying) (*Park et al., 2022*). Since human studies implicated the activity of the aIC to be more correlated with subjective fear experience (*Gogolla, 2017*; *Uddin et al., 2017*; *Terasawa et al., 2013*; *Torrence et al., 2019*), the aIC may be more specialized in regulating subjective fear and collectively integrating both anxiety and fear information. Although this study was unable to determine what determines an aIC neuron to be a freezing-excited or freezing-inhibited response type, we speculate that inputs from different brain areas may influence the different responses. For instance, freezing-inhibited cells may receive input from the prefrontal cortex since whole-brain connectivity analysis of the IC indicated that the prefrontal cortex provides more inputs to inhibitory neurons in the aIC than to excitatory neurons (*Gehrlach et al., 2020*). Conversely, freezing-excited cells may receive input from the amygdala, as the amygdala provides more input to excitatory neurons in the aIC than inhibitory neurons (*Gehrlach et al., 2020*).

Given the potential importance of freezing-excited and freezing-inhibited neurons in conditioned fear processing, we investigated whether disrupting the E/I balance by optogenetically increasing or decreasing the total output of aIC pyramidal neurons could alter fear behavior (freezing). Our findings demonstrated that fear behavior could be bidirectionally modulated by aIC neuronal activity: exciting aIC neurons increased the fear response, while inhibiting aIC neurons had the opposite

effect. These findings have significant clinical implications since they suggest that specifically targeting freezing-inhibited or freezing-excited cells for neuromodulation may not be necessary to have a therapeutic effect. Even though the aIC has fine microcircuitry and intricate connections with many brain regions, merely controlling the total output of the aIC may be enough to regulate fear and anxiety disorders.

During the aIC optogenetic stimulation, we observed a phenomenon where brief excitation or inhibition of aIC pyramidal neurons had a residual effect on behavior during the inter-trial period when optic stimulation was off. Another study also reports the cumulative effects of optical activation of ChR2-expressing neurons on freezing behavior even after optic stimulation ended (*Han et al., 2015*), supporting that it can happen. Although further studies will be necessary to verify what causes this phenomenon, our single unit recording and local field potential analysis during optostimulation support that it is not due to seizure produced by optostimulations. It may occur due to the recruitment of several brain areas modulated by the aIC neurons since activating a more specific circuit, the aIC→medial thalamus or the aIC→amygdala projection, did not produce any lasting effects between light stimulations. Activation or inhibition of the aIC may promote or reduce activation of a recurrent circuit that controls behavior, especially since the aIC has connections with motor-related brain regions (*Gehrlach et al., 2020*).

In addition to our finding that optically controlling the total output of the aIC bidirectionally and cumulatively regulates fear behavior, we found that selective activation of distinct non-overlapping population of aIC neurons projecting to different brain regions produced clear-cut bidirectional changes in fear behavior. Specifically, optical activation of the aIC projection to the amygdala enhanced and sustained fear behavior, while activation of the aIC projection to the thalamus reduced fear behavior. Our results are consistent with previous studies reporting that amygdala activation promotes fear responses (*Johansen et al., 2010*), while thalamus activation, especially in the burst firing mode of the MD, reduces fear responses (*Lee et al., 2011*; *Baek et al., 2019*). Although terminal optogenetic stimulation could produce antidromic activation (*Li et al., 2018*), it only activates a subset of aIC neurons projecting to the areas. Our study, therefore, demonstrates for the first time that controlling distinct aIC outputs bidirectionally regulate fear behaviors.

Although we do not have a direct link between single-unit recording and aIC projections, the recording locations of the freezing-excited and freezing-inhibited cells suggest that medial thalamus projecting neurons could be freezing-excited cells while the amygdala projecting neurons may be freezing-inhibited cells. If so, then the activity of the aIC neurons would work as a mechanism to reduce overexpression of fear. We would like to provide a more direct evidence between the neuronal response types and projection patterns in future studies by electrophysiologically identifying freezing-excited and freezing-inhibited aIC neurons and testing whether those neurons activates to optogenetic activation of amygdala or medial thalamus projecting aIC neurons. Similar to our findings, the insular→central amygdala and the insula→nucleus accumbens differentially regulated fear (*Wang et al., 2022*). Since the insular cortex is extensively connected with various brain regions (*Gehrlach et al., 2020*), several redundant circuits may be present in the aIC to fine-tune fear behavior. How the fear promoting and fear reducing aIC circuits compete and cooperate is still unclear and it would be interesting to investigate this in future studies.

Another factor to consider is that we have only used male mice in this study. Although many studies report that there is no biological sex difference in cued fear conditioning (*Day and Stevenson, 2020*), the main experimental paradigm used in this study, it does not mean that the underlying brain circuit mechanism would also be similar. The bidirectional fear modulation by aIC→medial thalamus or the aIC→amygdala projections may be different in female mice, as some studies report reduced cued fear extinction in females (*Day and Stevenson, 2020*).

In conclusion, our study provides evidence that the aIC regulates conditioned fear behaviors through bidirectional control on multiple levels, with the balance of freezing-excited and freezing-inhibited neuronal activity operating at the local circuit level and different populations of neurons projecting to different brain regions exerting additional control. These findings contribute to our understanding of the neural circuits underlying fear behavior and have potential implications for the development of novel therapeutic strategies for fear and anxiety disorders.

# Methods

## Animals

All experiments were conducted in accordance with the guidelines of the Institutional Animal Care and Use Committee (IACUC) of Ewha Womans University (EWHA IACUC 21–008 t), and all efforts were made to minimize animal suffering. Our research is in accordance with ARRIVE 2.0 guidelines (*Percie du Sert et al., 2020*). We used F1 hybrids of C57BL/6J×129S4/SvJaeJ male mice (10–18 weeks old; Jackson Laboratory, C57BL/6 J: 000664, 129S4/SvJaeJ: 009104) in our experiments. The mice were housed at a constant temperature (23 ± 1 °C) and humidity level (50 ± 5%). They were kept in home cages with free access to food and water and subjected to an alternating 12 hr dark-light reversal cycle starting at 9 am. Following the microdrive or optic ferrule implantation, the initially group-housed mice were individually caged. All experiments were carried out blinded to groups and the order of subjects in each experiment was randomized. Experiments were replicated at least twice and all data, unless otherwise indicated, were included in the analysis.

## Auditory fear conditioning

All mice were handled for at least one week before the experiments. For conditioning, we used a fear conditioning chamber (Coulbourn Instruments, Allentown, PA) equipped with stainless-steel bars on the floor for delivering electric foot shocks, placed inside an isolation box with a camera mounted on its ceiling. A day before conditioning, the mice were habituated in a white acrylic cylinder (context A: diameter * height = 20 * 25 cm) for 10 min. Twenty-four hours after habituation, the mice were exposed to three conditioning tones (CS; 2 kHz, 75 dB) lasting 30 s, which were co-terminated with a foot shock (US: 0.5 mA, 1 s) at 2.5 min inter-trial intervals in the fear conditioning chamber (context B; width * length * height = 18 * 18 * 29 cm). The expression of conditioned fear was tested the following day in context A with five CS lasting 30 s at 2.5 min intervals. For the fear extinction experiments, ten CS were presented at variable intervals (40–60 s; *Min et al., 2023*). The freezing response was defined as the absence of any movement (other than breathing) for >1 s and was videotaped and manually scored by two investigators who were blinded to the experimental groups. Freezing durations were summed in 30 s time bins.

## Surgery for single unit recording

Mice were anesthetized with Zoletil (30 mg/kg, i.p.) and then fixed to a stereotaxic instrument (David Kopf Instruments, USA) for surgical procedures. After craniotomy, a microdrive was implanted into the anterior insular cortex (coordinates from bregma: AP:+1.2 mm, ML: –3.4 mm, DV: –1.8 mm from brain surface, mouse brain atlas *Paxinos and Franklin, 2008*) for behavioral single-unit recordings. The microdrive was equipped with four tetrodes, where 12.5 μm Nichrome polyamide-insulated microwires were intertwined into one tetrode, Kanthal precision technology. The distance between the tetrodes were greater than 200 μm to ensure that distinct single-units will be obtained from different tetrodes. The recording tip of each tetrode channel was gold-plated to have a resistance of 300–400 kΩ measured at 1 kHz (Bak Electronics, USA). After implantation, the microdrive was secured onto the skull with self-tapping stainless-steel screws and dental cement (Vertex Dental, Netherlands). Mice were allowed to recover from surgery for at least one week before experiments.

## Single unit recording and expression of conditioned auditory fear

We used a behavioral single-unit recording technique to investigate the activities of individual neurons during exposure to the CS (tone) after auditory fear conditioning. Screening for neuronal activity was carried out in a square black chamber (width * length * height = 20 * 23 * 14 cm). To obtain unit signals, neural signals were amplified (x10,000), filtered (600 Hz to 6 kHz), and digitized (30.3 kHz) using Digital Lynx (Neuralynx, Tucson, AZ). Upon successful identification of unit signals, mice underwent habituation in context A, followed by the auditory fear conditioning process in context B the next day. No neuronal activities were recorded during this period. The day after conditioning, single neurons were recorded in freely behaving mice during the expression of auditory fear in context A. Neuronal activity and videos were recorded simultaneously and synchronized by the Neuralynx data acquisition system. Tone (CS; 2 kHz, 75 dB, lasting 30 s) event times were saved using Transistor-Transistor Logic (TTL) signals. The baseline was recorded for 3 min, and then behavior and neuronal activities during five CS (30 s), each separated by a 2.5 min inter-trial interval, were recorded. Neuronal

signals were manually isolated into single units using Spike Sort 3D (Neuralynx, USA). The quality of isolated unit signals was assessed by L-ratio, isolation distance, inter-spike intervals in the ISI histogram (ISI >1ms) provided by the software, and cross-correlation analysis. We only used well-isolated units (L-ratio <0.3, isolation distance >15) that were confirmed to be recorded in the aIC (conditioned group: n=116 neurons, 11 mice; control group: n=14 neurons, 3 mice) for the analysis (*Aoki et al., 2019*). The mean of units used in our analysis are as follows: L-ratio=0.09 ± 0.012, isolation distance = 44.97 ± 5.26 (expressed as mean ± standard deviation).

## Single unit recording analysis

Isolated single-unit signals were separated into putative pyramidal neurons (conditioned group: n=108 neurons, 11 mice; control group: n=13 neurons, 3 mice) or putative interneurons (conditioned group: n=8 neurons, 7 mice; control group: n=1 neuron, 1 mouse) using action potential duration and repolarization time (*Park et al., 2022*). The percentage of putative excitatory neurons and putative inhibitory interneurons obtained from both groups were similar (conditioned putative-excitatory: 93.1%, putative-inhibitory: 6.9%; control putative-excitatory: 92.9%, putative-inhibitory: 7.1%). We only used data from putative pyramidal neurons in our analysis. Firing rates (z-scored) of putative pyramidal neurons were calculated as follows: *value* = firing rate at a given time bin (*Figure 1G* and *Figure 2F*: 30 s, *Figure 2D*: 5 s, *Figure 2G*: 1 s), μ=mean firing rate of the total recording session, σ=standard deviation of the total recording session.

$$z = \frac{value - \mu}{\sigma}$$

For the tone-onset and offset analysis, firing rates (mean spikes/s) 200ms before and after the start of CS (tone-onset) and firing rates (mean spikes/s) 200ms before and after the end of CS (tone-offset) were analyzed in 10ms time bins. To determine whether firing rate changed significantly at the start or the end of CS, difference in firing rates in the presence or absence of CS were compared with the paired t-test for each cell. No cells significantly changed its firing rate at the start or the end of CS (significance determined at $p < 0.05$). Cell that did not have any spikes during the 400ms were excluded from the analysis (number of cells excluded from the analysis; freezing-inhibited: 1, freezing-excited: 10, non-responsive: 26, control: 5). For each group and cell-types, firing rates (mean ± standard deviation) around the start and end of CS were computed (*Figure 2H*).

The relationship between neuronal activity and behavior was analyzed by calculating Pearson's correlation coefficient between normalized firing rate (z-score) and freezing behavior analyzed in 30 s time bins. Based on the significance of Pearson's correlation coefficient (p-values <0.05), neurons were classified as freezing-excited cells (significant positive correlation), freezing-inhibited cells (significant negative correlation), or non-responsive cells (no significant correlation).

## Local field potential analysis

Local field potential (LFP) was simultaneously obtained with single unit recording. LFP frequency (1–120 Hz) changes induced by tone was analyzed with the *mtspecgramc* function from the Chronux toolbox (*Mitra and Bokil, 2007*). Normalized (z-scored) LFP power of the five tones before (20 s), during (30 s), and after (20 s) CS presentation were used to plot the spectrogram of the conditioned and control groups. To compare LFP power at different frequencies in presence and absence of tone in each group, a power spectrum was plotted with the Chronux toolbox's *mtspectrumc* function. Since previous studies report changes in theta power with fear conditioning in other brain regions (*Moita et al., 2003*; *Lesting et al., 2011*; *Buzsáki, 2002*), we analyzed the difference in theta (6 Hz) between groups in presence and absence of tone.

## Surgery for optogenetic studies

For the optogenetic studies, general anesthesia was induced in mice with 2.5% isoflurane (room air as the carrier gas) and maintained at 1.0% (Somnosuite, Kent Scientific) throughout surgical procedures. A 33-gauge blunt needle attached to a Nanofil (WPI, USA) was used for viral injections (500–600 nL/brain site). The injection speed was set at 100 nL/min, and injection needles were placed in the target site for 10 min before and after viral injection. Viruses with an alpha-calcium/calmodulin-dependent protein kinase II (CaMKIIα) promoter, which targets pyramidal neurons, were injected into the aIC

(coordinates from bregma: AP:+1.2 mm, ML: –3.4 mm, DV: –1.8 mm from brain surface) for optogenetic control of excitatory pyramidal neurons. The types of viruses used for the study were AAV2-CaMKIIα-ChR2-EYFP (n=13), AAV2-CaMKIIα-eNpHR3.0-EYFP (n=7), and AAV2-CaMKIIα-EYFP (n=13) (UNC Vector Core, USA). An optic fiber (multimode 62.5 µm core optic fiber) attached to a ceramic ferrule (Thorlabs, USA) was implanted unilaterally into the right aIC (AP:+1.2 mm, ML: –3.4 mm, DV: –1.75 mm). For stimulation of the aIC-amygdala or the aIC-medial thalamus projections, optic fibers were bilaterally implanted in the amygdala (AP: –1.2 mm, ML: ± 3.4 mm, DV: –4.25 mm) or the medial thalamus (AP: –1.2 mm, ML: ± 1.21 mm, DV: –2.53 mm, angle: ± 15°). Optic ferrules were secured onto the skull with self-tapping stainless-steel screws and dental cement (Vertex Dental, Netherlands). Mice for optogenetic experiments were allowed to recover for at least 2 weeks to allow sufficient time for viral expression.

### Open-field test with optogenetic stimulation

To assess whether optostimulation parameters used in our study may influence behavior and interfere with the interpretation of the results, we analyzed the behaviors of mice in the three groups (ChR2, NpHR3, control) with optostimulation recorded in an open field. Mice were released in an open field chamber (50 × 50 × 50 cm opaque white acrylic, ~75 lx) and allowed to explore for 3 min (baseline). Then, five 30 s optostimulations (light on) were delivered with an interval of 30 s (light off). The same optostimulation parameters used for the fear test was applied (ChR2: 20 Hz, 473 nm, 3–8 mW; NpHR3: continuous, 593 nm, 6–8 mW; control: 20 Hz, 473 nm, 3–8 mW). We used the Ethovision software to analyze the velocity (cm/s) of the three groups during baseline, light on, and light off.

### Neuronal recording with optogenetic stimulation

Changes in single unit activities and brain rhythms (LFP) induced by optostimulation were measured using a custom made optotetrode (microdrive with an optic fiber and 4 tetrodes, Axona, UK) in freely moving mice. The same single unit recording and analysis, and optostimulation parameters were used (see according sections). To obtain LFP signals, neuronal signals were bandpass filtered (0.1 Hz to 8 kHz) and digitized (32 kHz). We inspected possible seizure-like events (ictal activity) generated by optogenetic stimulations with raw traces of LFP and did not find any. Changes in brain rhythm induced by optogenetic stimulation were analyzed by plotting time-frequency spectrograms calculated using the *mtspecgramc* function in the Chronux toolbox (*Mitra and Bokil, 2007*).

### Fear expression with optogenetic stimulation

We investigated how optically activating aIC neurons or activating specific output targets of the aIC—either to the amygdala or medial thalamus—modulates conditioned fear. On the day following conditioning, the fear expression of mice was measured with continuous optical stimulation delivered (ChR2: 473 nm, 3–8 mW; NpHR3: 593 nm, 6–8 mW; eYFP: 473 nm, 3–8 mW) during each CS presentation in context A. The fear expression paradigm for the aIC opto-stimulation was the same as the one for the single unit recording study. For the aIC terminal stimulation, bilateral optical stimulation (20 Hz pulses, 473 nm, 1–2 mW) was delivered during CS presentation in either the medial thalamus or the amygdala. Ten CSs were presented with variable inter-CS intervals (40–60 s). The behaviors of mice in each group were videotaped and scored as described above.

### Tracer injection surgery

To investigate the distribution of neurons that project to either the amygdala or the medial thalamus, two retrograde tracers (CTB; Cholera toxin B) conjugated with different fluorophores were injected into a mouse. Retrograde tracers were injected into the amygdala (400 nL of 1% Alexa Fluor 488-CTB, AP: –1.2 mm, ML: –3.4 mm, DV: –4.35 mm) and the medial thalamus (400 nL of 1% Alexa Fluor 594-CTB, AP: –1.2 mm, ML: –0.2 mm, DV: –3.4 mm) in each mouse (n=3). The same surgical and injection protocols used for the viral injection in the optogenetic experiments were used.

### Histology

After completion of the study, mice were overdosed with 2% avertin. To locate tetrode tips, a small electrolytic lesion was made at the tip of the recording site by passing an anodic current (10 µA, 5 s) through one tetrode channel before the perfusion. Anesthetized mice were transcardially perfused

with saline (0.9%) followed by 10% buffered neutral formalin. Brains were removed and stored in 10% formalin for a day, then transferred to a 30% sucrose solution for cryoprotection. Fixed brain tissues were cut and frozen with a microtome (HM525 NX, Thermo Fisher) in coronal sections (40 µm). To locate tetrode tips, brain sections were stained with cresyl violet (Sigma, USA) and examined under a light microscope (Axioscope 5, Carl Zeiss, Germany) to locate lesions. To assess viral vector expression, sectioned brain slices were examined with a fluorescent microscope (Axioscope 5, Carl Zeiss, Germany) with a CoolLED pE 300 series illumination system. The fluorescence of retrograde tracers was imaged using a laser-scanning confocal microscope (LSM 700, Carl Zeiss, Germany).

### Statistical analysis

All statistical analyses were performed using SPSS 26.0 (SPSS Inc, USA). For normally distributed samples, an unpaired two-tailed t-test or one-way ANOVA followed by Bonferroni was used to compare means between groups. A binomial proportions test was used to compare the difference between the proportions of the freezing-excited and freezing-inhibited groups. To compare the means of two groups with unequal variance, the Mann-Whitney U test was used. A Kruskal-Wallis test followed by Dunn's post hoc analysis was used for comparison among three groups with unequal variance. To compare changes that occur over time, within a group or between groups, repeated measures ANOVA followed by Bonferroni was used. For within group comparisons paired t-test was used. To test the presence of interaction between the opsin and optostimulation effect, two-way ANOVA followed by Bonferroni post hoc was used.

### Acknowledgements

This work was supported by the National Research Foundation of Korea (NRF) grant funded by the Ministry of Science and ICT [NRF-2021R1A6A1A10039823(SG), NRF-2021R1C1C1006607 (YH), NRF-2022M3E5E8018421 (JC), and NRF-2022R1A2C2009265 (JC)] and by NIH grant MH099073 (JJK).

## Additional information

### Funding

| Funder | Grant reference number | Author |
|---|---|---|
| National Research Foundation of Korea | 2021R1A6A1A10039823 | Sanggeon Park |
| National Research Foundation of Korea | 2021R1C1C1006607 | Yeowool Huh |
| National Research Foundation of Korea | 2022M3E5E8018421 | Jeiwon Cho |
| National Research Foundation of Korea | 2022R1A2C2009265 | Jeiwon Cho |
| National Institute of Mental Health | MH099073 | Jeansok J Kim |

The funders had no role in study design, data collection and interpretation, or the decision to submit the work for publication.

### Author contributions

Sanggeon Park, Yeowool Huh, Conceptualization, Resources, Data curation, Software, Formal analysis, Funding acquisition, Validation, Investigation, Visualization, Methodology, Writing – original draft, Writing – review and editing; Jeansok J Kim, Conceptualization, Data curation, Formal analysis, Supervision, Funding acquisition, Validation, Visualization, Methodology, Writing – original draft, Writing – review and editing; Jeiwon Cho, Conceptualization, Data curation, Formal analysis, Supervision, Funding acquisition, Validation, Visualization, Methodology, Writing – original draft, Project administration, Writing – review and editing

## Author ORCIDs

Sanggeon Park (ID) https://orcid.org/0000-0003-2083-2536
Jeansok J Kim (ID) https://orcid.org/0000-0001-7964-106X
Jeiwon Cho (ID) https://orcid.org/0000-0001-6903-3562

## Ethics

All experiments were conducted in accordance with the guidelines of the Institutional Animal Care and Use Committee (IACUC) of Ewha Womans University (EWHA IACUC 21-008-t), and all efforts were made to minimize animal suffering. Our research is in accordance with ARRIVE 2.0 guidelines.

Reviewer #1 (Public review): https://doi.org/10.7554/eLife.95821.3.sa1
Reviewer #2 (Public review): https://doi.org/10.7554/eLife.95821.3.sa2
Author response https://doi.org/10.7554/eLife.95821.3.sa3

# Additional files

## Supplementary files

• MDAR checklist

## Data availability

All data used in the figures have been deposited at Open Science Framework (https://osf.io/ubyms).

The following dataset was generated:

| Author(s) | Year | Dataset title | Dataset URL | Database and Identifier |
|---|---|---|---|---|
| Park S | 2024 | Bidirectional fear modulation by discrete anterior insular circuits in male mice | https://osf.io/ubyms | Open Science Framework, ubyms |

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
