## [Editor Report · eLife assessment]

This work provides a **valuable** characterization of neural activity in the anterior insular cortex during fear. Using behavior, single unit recording, and optogenetic control of neural activity, the paper provides **convincing** data on the role of anterior insular circuits in bidirectionally controlling fear. The study is a great starting point on the path to testing hypotheses about bidirectional control of behavior via neural activity in anatomically defined output populations.

---

## [Referee Report · Reviewer #1 (Public review)]

The authors tested whether anterior insular cortex neurons that increase or decrease firing during fear behavior, freezing, bidirectionally control fear via separate, anatomically defined outputs. Using a fairly simple behavior where mice were exposed to tone-shock pairings, they found roughly equal populations that increased or decreased firing during freezing. They next tested whether these distinct populations also had distinct outputs. Using retrograde tracers they found that the anterior insular cortex contains non-overlapping neurons which project to the mediodorsal thalamus or amygdala. Mediodorsal thalamus-projecting neurons tended to cluster in deep cortical layers, while amygdala-projecting neurons were primarily in more superficial layers. Stimulation of insula-thalamus projection decreased freezing behavior, and stimulation of insula-amygdala projections increased fear behavior. Given that the neurons which increased firing were located in deep layers, that thalamus projections occurred in deep layers, and that stimulation of insula-thalamus neurons decreased freezing, the authors concluded that the increased-firing neurons were likely thalamic projections. Similarly, given that decreased-firing neurons tended to occur in more superficial layers, that insula-amygdala projections were primarily superficial, and that insula-amygdala stimulation increased freezing behavior, authors concluded that the decreased firing cells were likely amygdala projections. The study has several strengths though also some caveats. Overall, the authors provide a valuable contribution to the field by demonstrating bidirectional control of behavior, linking the underlying anatomy and physiology.

Strengths:

The potential link between physiological activity, anatomy, and behavior is well laid out and is an interesting question. The activity contrast between the units that increase/decrease firing during freezing is clear.

It is nice to see the recording of extracellular spiking activity, which provides a clear measure of neural output, whereas similar studies often use bulk calcium imaging, a signal which rarely matches real neural activity even when anatomy suggests it might.

Weaknesses:

The link between spiking, anatomy, and behavior requires assumptions/inferences: the anatomically/genetically defined neurons which had distinct outputs and opposite behavioral effects can only be assumed the increased/decreased spiking neurons, based on the rough area of cortical layer they were recorded. This is, of course, discussed as a future experiment.

---

## [Referee Report · Reviewer #2 (Public review)]

In this study, the authors aim to understand how neurons in the anterior insular cortex (insula) modulate fear behaviors. They report that the activity of a subpopulation of insula neurons is positively correlated with freezing behaviors, while the activity of another subpopulation of neurons is negatively correlated to the same freezing episodes. They then used optogenetics and showed that activation of anterior insula excitatory neurons during tones predicting a footshock increases the amount of freezing outside the tone presentation, while optogenetic inhibition had no effect. Finally, they found that two neuronal projections of the anterior insula, one to the amygdala and another to the medial thalamus, are increasing and decreasing freezing behaviors respectively.

---

## [Author Response]

The following is the authors’ response to the original reviews.

**Reviewer #1 (Public Review):**
The authors sought to test whether anterior insular cortex neurons increase or decrease firing during fear behavior and freezing, bi-directionally control fear via separate, anatomically defined outputs. Using a fairly simple behavior where mice were exposed to tone-shock pairings, they found roughly equal populations that do indeed either increase or decrease firing during freezing. Next, they sought to test whether these distinct populations may also have distinct outputs. Using retrograde tracers they found that the anterior insular cortex contains non-overlapping neurons which project to the mediodorsal thalamus or amygdala. Mediodorsal thalamus-projecting neurons tended to cluster in deep cortical layers while amygdala-projecting neurons were primarily in more superficial layers. Stimulation of insula-thalamus projection decreased freezing behavior, and stimulation of insula-amygdala projections increased fear behavior. Given that the neurons that increased firing were located in deep layers, that thalamus projections occurred in deep layers, and that stimulation of insula-thalamus neurons decreased freezing, the authors concluded that the increased firing neurons may be thalamus projections. Similarly, given that decreased-firing neurons tended to occur in more superficial layers, that insula-amygdala projections were primarily superficial, and that insula-amygdala stimulation increased freezing behavior, authors concluded that the decreased firing cells may be amygdala projections. The study has several strengths though also some caveats.Strengths:The potential link between physiological activity, anatomy, and behavior is well laid out and is an interesting question. The activity contrast between the units that increase/decrease firing during freezing is clear.It is nice to see the recording of extracellular spiking activity, which provides a clear measure of neural output, whereas similar studies often use bulk calcium imaging, a signal that rarely matches real neural activity even when anatomy suggests it might (see London et al 2018 J Neuro - there are increased/decreased spiking striatal populations, but both D1 and D2 striatal neurons increase bulk calcium).Weaknesses:The link between spiking, anatomy, and behavior requires assumptions/inferences: the anatomically/genetically defined neurons which had distinct outputs and opposite behavioral effects can only be assumed the increased/decreased spiking neurons, based on the rough area of the cortical layer they were recorded.

Yes, we are aware that we could not provide a direct link between spiking, anatomy and behavior. We have specifically noted this in the discussion section and added a possible experiment that could be carried out to provide a more direct link in a future study.

[Lines 371-375] We would like to provide a more direct evidence between the neuronal response types and projection patterns in future studies by electrophysiologically identifying freezing-excited and freezing-inhibited aIC neurons and testing whether those neurons activates to optogenetic activation of amygdala or medial thalamus projecting aIC neurons.

The behavior would require more control to fully support claims about the associative nature of the fear response (see Trott et al 2022 eLife) - freezing, in this case, could just as well be nonassociative. In a similar vein, fixed intertrial intervals, though common practice in the fear literature, pose a problem for neurophysiological studies. The first is that animals learn the timing of events, and the second is that neural activity is dynamic and changes over time. Thus it is very difficult to determine whether changes in neural activity are due to learning about the tone-shock contingency, timing of the task, simply occur because of time and independently of external events, or some combination of the above.

Trott et al. (2022) stated that "...freezing was the purest reflection of associative learning." The nonassociative processes mentioned in the study were related to running and darting behaviors, which the authors argue are suppressed by associative learning. Moreover, considerable evidence from immediate postshock freezing and immediate postshock context shift studies all indicate that the freezing response is an associative (and not nonassociative) response (Fanselow, 1980 and 1986; and Landeira-Fernandez et al., 2006). Thus, our animals' freezing response to the tone CS presentation in a novel context, following three tone CS-footshock US pairings, most likely reflects associative learning.

Concerning the issue of fixed inter-trial intervals (ITIs), which are standard in fear conditioning studies, particularly those with few CS-US paired trials, we acknowledge the challenge in interpreting the neural correlates of behavior. However, the ITIs in our extinction study was variable and we still found neural activities that had significant correlation with freezing. The results of our extinction study, carried out with variable it is, suggest that the aIC neural activity changes measured in this study is likely due to freezing behavior associated with fear learning, not due to learning the contingencies of fixed ITIs.

**Reviewer #2 (Public Review):**
In this study, the authors aim to understand how neurons in the anterior insular cortex (insula) modulate fear behaviors. They report that the activity of a subpopulation of insula neurons is positively correlated with freezing behaviors, while the activity of another subpopulation of neurons is negatively correlated to the same freezing episodes. They then used optogenetics and showed that activation of anterior insula excitatory neurons during tones predicting a footshock increases the amount of freezing outside the tone presentation, while optogenetic inhibition had no effect. Finally, they found that two neuronal projections of the anterior insula, one to the amygdala and another to the medial thalamus, are increasing and decreasing freezing behaviors respectively. While the study contains interesting and timely findings for our understanding of the mechanisms underlying fear, some points remain to be addressed.

We are thankful for the detailed and constructive comments by the reviewer and addressed the points. Specifically, we included possible limitations of using only male mice in the study, included two more studies about the insula as references, specified the L-ratio and isolated distance used in our study, added the ratio of putative-excitatory and putative-inhibitory neurons obtained from our study, changed the terms used to describe neuronal activity changes (freezing-excited and freezing-inhibited cells), added new analysis (Figure 2H), rearranged Figure 2 for clarity, added new histology images, and added atlas maps with viral expressions (three figure supplements).

**Reviewer #1 (Recommendations For The Authors):**
- I would suggest keeping the same y-axis for all figures that display the same data type - Figure 5D, for example.

Thank you for the detailed suggestion. We corrected the y-axis that display the same data type to be the same for all figures.

- In the methods, it says 30s bins were used for neural analysis (line 435). I cannot imagine doing this, and looking at the other figures, it does not look like this is the case so could you please clarify what bins, averages, etc were used for neural and behavioral analysis?

Bin size for neural analysis varied; 30s, 5s, 1s bins were used depending on the analysis. We corrected this and specified what time bin was used for which figure in the methods.

Bin size for neural and freezing behavior was 30s and we also added this to the methods.

- I would not make any claims about the fear response here being associative/conditional. This would require a control group that received an equal number of tone and shock exposures, whether explicitly unpaired or random.

The unpaired fear conditioning paradigm, unpaired tone and shock, suggested by the reviewer is well characterized not to induce fear behavior by CS (Moita et al., 2003 and Kochli et al., 2015). In addition, considerable evidence from immediate post-shock freezing and immediate post-shock context shift studies all indicate that the freezing response is an associative (and not nonassociative) response (Fanselow, 1980 and 1986; and Landeira-Fernandez et al., 2006). Thus, our animals' freezing response to the tone CS presentation in a novel context, following three tone CS-footshock US pairings, most likely reflects associative learning.

- I appreciate the discussion about requiring some inference to conclude that anatomically defined neurons are the physiologically defined ones. This is a caveat that is fully disclosed, however, I might suggest adding to the discussion that future experiments could address this by tagging insula-thalamus or insula-amygdala neurons with antidromic (opto or even plain old electric!) stimulation. These experiments are tricky to perform, of course, but this would be required to fully close all the links between behavior, physiology, and anatomy.

As suggested, we have included that, in a future study, we would like to elucidate a more direct link between physiology, anatomy and behaviors by optogenetically tagging the insula-thalamus/insula-amygdala neurons and identifying whether it may be a positive or a negative cell (now named the freezing-excited and freezing-inhibited cells, respectively) in the discussion.

[Lines 371-375] We would like to provide a more direct evidence between the neuronal response types and projection patterns in future studies by electrophysiologically identifying freezing-excited and freezing-inhibited aIC neurons and testing whether those neurons activates to optogenetic activation of amygdala or medial thalamus projecting aIC neurons.

**Reviewer #2 (Recommendations For The Authors):**
Major comments:(1) As all experiments have been performed only in male mice, the authors need to clearly state this limit in the introduction, abstract, and title of the manuscript.

With increasing number of readers becoming interested in the biological sex used in preclinical studies, we also feel that it should be mentioned in the beginning of the manuscript. As suggested, we explicitly wrote that we only used male mice in the title, abstract, and introduction. In addition, we discussed possible limitations of only using male mice in the discussion section as follows:

[Lines 381-386] Another factor to consider is that we have only used male mice in this study. Although many studies report that there is no biological sex difference in cued fear conditioning (42), the main experimental paradigm used in this study, it does not mean that the underlying brain circuit mechanism would also be similar. The bidirectional fear modulation by aIC→medial thalamus or the aIC→amygdala projections may be different in female mice, as some studies report reduced cued fear extinction in females (42).

(2) The authors are missing important publications reporting findings on the insular cortex in fear and anxiety. For example, the authors should cite studies showing that anterior insula VIP+ interneurons inhibition reduces fear memory retrieval (Ramos-Prats et al., 2022) and that posterior insula neurons are a state-dependent regulator of fear (Klein et al., 2021). Also, regarding the anterior insula to basolateral amygdala projection (aIC-BLA), the author should include recent work showing that this population encodes both negative valence and anxiogenic spaces (Nicolas et al., 2023).

We appreciate the detailed suggestions and we added appropriate publications in the discussion section. The anterior insula VIP+ interneuron study (Ramos-Prats et al., 2022) is interesting, but based on the evidence provided in the paper, we felt that the role of aIC VIP+ interneuron in fear conditioning is low. VIP+ interneurons in the aIC seem to be important in coding sensory stimuli, however, it’s relevance to conditioned stimuli seems to be low; overall VIP intracellular calcium activity to CS was low and did not differ between acquisition and retrieval. Also, inhibition of VIP did not influence fear acquisition. VIP inhibition during fear acquisition did reduce fear retrieval (CS only, no light stimulation), but this does not necessarily mean that VIP activity will be involved in fear memory storage or retrieval, especially because intracellular calcium activity of VIP+ neurons was low during fear conditioning and retrieval.

Studies by Klein et al. (2021) and Nicolas et al. (2023) are integrated in the discussion section as follows.

[Lines 297-301] Group activity of neurons in the pIC measured with fiberphotometry, interestingly, exhibited fear state dependent activity changes—decreased activity with high fear behavior and increased activity with lower fear behavior (29)—suggesting that group activity of the pIC may be involves in maintain appropriate level of fear behavior.

[Lines 316-319] Another distinction between the aIC and pIC may be related with anxiety, as a recent study showed that group activity of aIC neurons, but not that of the pIC, increased when mice explored anxiogenic space (open arms in an elevated plus maze, center of an open field box) (32).

(3) The authors should specify how many neurons they excluded after controlling the L-ratio and isolation distance. It is also important to specify the percentage of putative excitatory and inhibitory interneurons recorded among the 11 mice based on their classification (the number of putative inhibitory interneurons in Figure 1D seems too low to be accurate).

We use manual cluster cutting and only cut clusters that are visually well isolated. So we hardly have any neurons that are excluded after controlling for L-ratio and isolation distance. The criterion we used was L-ratio<0.3 and isolation distance>15, and we specified this in the methods as follows.

[Lines 454-458] We only used well-isolated units (L-ratio<0.3, isolation distance>15) that were confirmed to be recorded in the aIC (conditioned group: n = 116 neurons, 11 mice; control group: n = 14 neurons, 3 mice) for the analysis (46). The mean of units used in our analysis are as follows: L-ratio = 0.09 ± 0.012, isolation distance = 44.97 ± 5.26 (expressed as mean ± standard deviation).

As suggested, we also specified the percentage of putative excitatory and inhibitory interneurons recorded from our study in the results and methods section. The relative percentage of putative excitatory and inhibitory interneurons were similar for both the conditioned and the control groups (conditioned putative-excitatory: 93.1%, putative-inhibitory: 6.9%; control putative-excitatory: 92.9%, putative-inhibitory: 7.1%). Although the number of putative-interneurons isolated from our recordings is low that is what we obtained. Putative inhibitory neurons, probably because of their relatively smaller size, has a tendency to be underrepresented than the putative excitatory cells.

[Lines 83-87] Of the recorded neurons, we analyzed the activity of 108 putative pyramidal neurons (93% of total isolated neurons) from 11 mice, which were distinguished from putative interneurons (n = 8 cells, 7% of total isolated neurons) based on the characteristics of their recorded action potentials (Figure 1D; see methods for details).

[Lines 464-467] The percentage of putative excitatory neurons and putative inhibitory interneurons obtained from both groups were similar (conditioned putative-excitatory: 93.1%, putative-inhibitory: 6.9%; control putative-excitatory: 92.9%, putative-inhibitory: 7.1%).

(4) While the use of correlation of single-unit firing frequency with freezing is interesting, classically, studies analyze the firing in comparison to the auditory cues. If the authors want to keep the correlation analysis with freezing, rather than correlations to the cues, they should rename the cells as "freezing excited" and "freezing inhibited" cells instead of positive and negative cells.

As suggested, we used the terms “freezing-excited” and “freezing-inhibited” cells instead of positive and negative cells.

(5) To improve clarity, Figure 2 should be reorganized to start with the representative examples before including the average of population data. Thus Panel D should be the first one. The authors should also consider including the trace of the firing rate of these representative units over time, on top of the freezing trace, as well as Pearson's r and p values for both of them. Then, the next panels should be ordered as follows: F, G, H, C, A, B, I, and finally E.

We have rearranged Figure 2 based on the suggestions.

(6) It is unclear why the freezing response in Figure 2 is different in current panels F, G, and H. Please clarify this point.

It was because the freezing behaviors of slightly different population of animals were averaged. Some animals did not have positive/negative (or both) cells and only the behavior of animals with the specified cell-type were used for calculating the mean freezing response. With rearrangement of Figure 2, now we do not have plots with juxtaposed mean neuronal response-types and behavior.

(7) Even though the peak of tone-induced firing rate change between negative and positive cells is 10s later for positive cells, the conclusion that this 'difference suggests differential circuits may regulate the activities of different neuron types in response to fear' is overstating the observation. This statement should be rephrased. Indeed, it could be the same circuits that are regulated by different inputs (glutamatergic, GABA, or neuromodulatory inputs).

We agree and delete the statement from the manuscript.

(8) The authors mention they did not find tone onset nor tone offset-induced responses of anterior insula neurons. It would be helpful to represent this finding in a Figure, especially, which were the criteria for a cell to be tone onset or tone offset responding.

We added how tone-onset and tone-offset were analyzed in the methods section and added a plot of the analysis in Figure 2H.

(9) Based on the spread of the viral expression shown in Figure 3B, it appears that the authors are activating/inhibiting insula neurons in the GI layer, whereas single-unit recordings report the electrodes were located in DI, AID, and AIV layers. The authors should provide histology maps of the viral spread for ChR2, NpHR3, and eYFP expression.

Thank you for the excellent suggestion. Now the histological sample in Figure 3B is a sample with expression in the GI/DI/AID layers and it also has an image taken at higher resolution (x40) to show that viral vectors are expressed inside neurons. We also added histological maps with overlay of viral expression patterns of the ChR2, eYFP, and NpHR3 groups in Figure 3—figure supplement 1.

(10) In Figure 5B, the distribution of terminals expressing ChR2 appears much denser in CM than in MD. This should be quantified across mice and if consistent with the representative image, the authors should refer to aIC-CM rather than aIC-MD terminals.

Overall, we referred to the connection as aIC-medial thalamus, which collectively includes both the CM and the MD. Microscopes we have cannot determine whether terminals end at the CM or MD, but the aIC projections seems to pass through the CM to reach the MD. The Allen Brain Institute’s Mouse brain connectivity map (https://connectivity.brain-map.org/projection/experiment/272737914) of a B6 mouse, the mouse strain we used in our study, with tracers injected in similar location as our study also supports our speculation and shows that aIC neuronal projections terminate more in the MD than in the CM. In addition, the power of light delivered for optogenetic manipulation is greatly reduced over distance, and therefore, the MD projecting terminals which is closer to the optic fiber will be more likely to be activated than the CM projecting terminals. However, since we could not determine whether the aIC terminate at the CM or the MD, we collectively referred to the connection as the aIC-medial thalamus throughout the manuscript.

(11) Histological verifications for each in vivo electrophysiology, optogenetic, and tracing experiments need to include a representative image of the implantation/injection site, as well as a 40x zoom-in image focusing on the cell bodies or terminals right below the optic fiber (for optogenetic experiments). Moreover, an atlas map including all injection locations with the spread of the virus and fiber placement should be added in the Supplement Figures for each experiment (see Figure S1 Klein et al., 2021). Similarly, the authors need to add a representation of the spread of the retrograde tracers for each mouse used for this tracing experiment.

As suggested, we added a histology sample showing electrode recording location for in-vivo electrophysiology in Figure 1 and added atlas maps for the optogenetic and tracing experiments in supplementary figures. We also provide a 40x zoom-in image of the expression pattern for the optogenetic experiments (Figure 3B).

(12) To target anterior insula neurons, authors mention coordinates that do not reach the insula on the Paxinos atlas (AP: +1.2 mm, ML: -3.4 mm, DV: -1.8 mm). If the DV was taken from the brain surface, this has to be specified, and if the other coordinates are from Bregma, this also needs to be specified. Finally, the authors cite a review from Maren & Fanselow (1996), for the anterior insula coordinates, but it remains unclear why.

AP and ML coordinates are measurement made in reference to the bregma. DV was calculated from the brain surface. We specified these in the Methods. We did not cite a review from Maren & Fenselow for the aIC coordinates.

Minor comments:

(1) A schematic of the microdrive and tetrodes, including the distance of each tetrode would also be helpful.

We used a handcrafted Microdrives with four tetrodes. Since they were handcrafted, the relative orientation of the tetrodes varies and tetrode recording locations has to be verified histologically. We, however, made sure that the distance between tetrodes to be more than 200 μm apart so that distinct single-units will be obtained from different tetrodes. We added this to the methods as follows.

[Lines 430-431] The distance between the tetrodes were greater than 200 μm to ensure that distinct single-units will be obtained from different tetrodes.

(2) Figure 2E: representation of the baseline firing (3-min period before the tone presentation) is missing.

Figure 2E is the 3 min period before tone presentation

(3) Figure 2: Averages Pearson's correlation r and p values should be stated on panels F, G, and H (positive cell r = 0.81, P < 0.05; negative cell r = -0.68, P < 0.05).

They were all originally stated in the figures. But with reorganization of Figure 2, we now have a plot of the Pearson’s Correlation with r and p values in Figure 2F.

(4) Figure 2I: Representation of the absolute value of the normalized firing is highly confusing. Indeed, as the 'negative cells' are inhibited to freezing, firing should be represented as normalized, and negative for the inhibited cells.

To avoid confusion, we did not take an absolute value of the “negative cells”, which are now called the “freezing-inhibited cells”.

(5) Figure 4E (retrograde tracing): representation of individual values is missing.

Figure 4E now has individual values.

References:

London, T. D., Licholai, J. A., Szczot, I., Ali, M. A., LeBlanc, K. H., Fobbs, W. C., & Kravitz, A. V. (2018). Coordinated ramping of dorsal striatal pathways preceding food approach and consumption. Journal of Neuroscience, 38(14), 3547-3558.

Trott, J. M., Hoffman, A. N., Zhuravka, I., & Fanselow, M. S. (2022). Conditional and unconditional components of aversively motivated freezing, flight and darting in mice. Elife, 11, e75663.

Fanselow, M. S. (1980). Conditional and unconditional components of post-shock freezing. The Pavlovian journal of biological science: Official Journal of the Pavlovian, 15(4), 177-182.

Fanselow, M. S. (1986). Associative vs topographical accounts of the immediate shock-freezing deficit in rats: implications for the response selection rules governing species-specific defensive reactions. Learning and Motivation, 17(1), 16-39.

Landeira-Fernandez, J., DeCola, J. P., Kim, J. J., & Fanselow, M. S. (2006). Immediate shock deficit in fear conditioning: effects of shock manipulations. Behavioral neuroscience, 120(4), 873.

Moita, M. A., Rosis, S., Zhou, Y., LeDoux, J. E., & Blair, H. T. (2003). Hippocampal place cells acquire location-specific responses to the conditioned stimulus during auditory fear conditioning. Neuron, 37(3), 485-497.

Kochli, D. E., Thompson, E. C., Fricke, E. A., Postle, A. F., & Quinn, J. J. (2015). The amygdala is critical for trace, delay, and contextual fear conditioning. Learning & memory, 22(2), 92-100.

Ramos-Prats, A., Paradiso, E., Castaldi, F., Sadeghi, M., Mir, M. Y., Hörtnagl, H., ... & Ferraguti, F. (2022). VIP-expressing interneurons in the anterior insular cortex contribute to sensory processing to regulate adaptive behavior. Cell Reports, 39(9).

Klein, A. S., Dolensek, N., Weiand, C., & Gogolla, N. (2021). Fear balance is maintained by bodily feedback to the insular cortex in mice. Science, 374(6570), 1010-1015.

Nicolas, C., Ju, A., Wu, Y., Eldirdiri, H., Delcasso, S., Couderc, Y., ... & Beyeler, A. (2023). Linking emotional valence and anxiety in a mouse insula-amygdala circuit. Nature Communications, 14(1), 5073.

Maren, S., & Fanselow, M. S. (1996). The amygdala and fear conditioning : Has the nut been cracked? Neuron, 16(2), 237‑240. https://doi.org/10.1016/s0896-6273(00)80041-0